# Exploring Adversarial Robustness of Multi-sensor Perception Systems in Self Driving

[1,2] **James Tu**   [3] **Huichen Li**   **Xinchen Yan**   [1,2] **Mengye Ren**   [1,2] **Yun Chen**
[1] **Ming Liang**   [4] **Eilyan Bitar**   **Ersin Yumer**   [1,2] **Raquel Urtasun**

Waabi[1], University of Toronto[2], UIUC[3], Cornell University[4]
`{jtu, mren, yun, urtasun}@cs.toronto.edu`
`huichen3@illinois.edu   eyb5@cornell.edu`
`{skywalkeryxc,liangming.elgoog,meyumer}@gmail.com`

**Abstract:** Modern self-driving perception systems have been shown to improve upon processing complementary inputs such as LiDAR with images. In isolation, 2D images have been found to be extremely vulnerable to adversarial attacks. Yet, there are limited studies on the adversarial robustness of multi-modal models that fuse LiDAR and image features. Furthermore, existing works do not consider physically realizable perturbations that are consistent across the input modalities. In this paper, we showcase practical susceptibilities of multi-sensor detection by inserting an adversarial object on a host vehicle. We focus on physically realizable and input-agnostic attacks that are feasible to execute in practice, and show that a single universal adversary can hide different host vehicles from state-of-the-art multi-modal detectors. Our experiments demonstrate that successful attacks are primarily caused by easily corrupted image features. Furthermore, in modern sensor fusion methods which project image features into 3D, adversarial attacks can exploit the projection process to generate false positives in distant regions in 3D. Towards more robust multi-modal perception systems, we show that adversarial training with feature denoising can boost robustness to such attacks significantly.

**Keywords:** Adversarial, Self-Driving, Perception, Multimodal

## 1   Introduction

Recent advances in self-driving perception have shown that fusing information from multiple sensors (e.g., camera, LiDAR, radar) [1, 2, 3, 4, 5, 6] leads to superior performance when compared to relying on single sensory inputs. Such performance gains are primarily due to the complementary information contained in the measurements provided by the different types of sensors. For example, LiDAR sensors provide accurate 3D geometry while cameras capture rich appearance information.

Meanwhile, modern perception models which rely on deep neural networks (DNNs) have been found to be extremely vulnerable to adversarial attacks when processing images in isolation [7, 8, 9, 10, 11]. Adversarial attacks can be thought of as perturbations to the sensory inputs which do not alter the semantic meaning of the scene, but drastically change a DNN's output and resulting in incorrect predictions. Such vulnerabilities can lead to catastrophic consequences in safety-critical applications. In the context of self-driving, most efforts have investigated attacks against single-sensor inputs, such as image-only attacks [7, 11] and LiDAR-only attacks [12]. Towards multi-modal robustness, [13] considers perturbations of LiDAR and image inputs independently, resulting in perturbations that are inconsistent across modalities and therefore may not be physically realizable and hence not threatening in practice. On the other hand, some proposed physically realizable approaches [14] only search over shape but ignore texture which is crucial for corrupting image inputs. Furthermore, these prior works do not generate universal perturbations which are perhaps the most threatening in practice. Such perturbations are input agnostic and can attack any input in the training distribution with high probability, meaning they can be executed without prior knowledge of the scene and are able to consistently disrupt models that process sensory information across time.

---

Work done while all authors were at UberATG.

5th Conference on Robot Learning (CoRL 2021), London, UK.

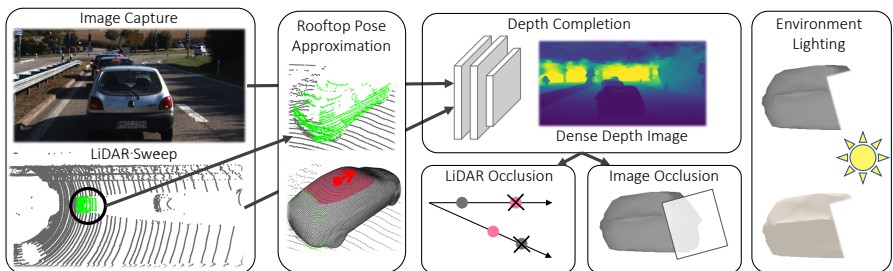

Figure 1: Simulating the addition of a mesh onto a vehicle rooftop in a realistic manner. First roof approximation is done to determine placement location and heading. Then LiDAR points and pixels are rendered with directional lighting to approximate sunlight. Finally, a dense depth image is generated with depth completion and used to handle occlusion.

This paper demonstrates the susceptibility of multi-sensor detection models to physically realizable and input-agnostic adversarial perturbations. To create a physically realizable attack which is also feasible to execute, we focus on object insertion attacks [7, 15, 16, 17, 12], as they can be carried out via the deployment of physical objects in the real world. Following [12], we insert the adversarial object by placing it on the rooftop of a host vehicle. We render the adversary into LiDAR and image inputs to ensure perturbations are consistent across modalities and that our attack is physically realizable. Furthermore, we consider occlusion and environmental lighting in the rendering process as shown in Figure 1 to enhance the realism of simulation. To enable end-to-end learning of geometry and texture, we render the pixels and LiDAR points in a differentiable manner. During training, our adversary is optimized with respect to all vehicles in the training distribution to create a universal attack which can be applied to any vehicle in any scene.

We conduct an empirical evaluation of our proposed attack on the KITTI [18] self-driving dataset and a novel large-scale self-driving dataset XENITH using the multi-sensor detector MMF [5]. We generate input-agnostic adversarial examples that successfully hide host vehicles from state-of-the-art detectors in both datasets. More importantly, we find that incorporating image inputs makes models more vulnerable when compared to using LiDAR alone, as successful attacks are primarily caused by the brittle image features. Moreover, the projection of image features into 3D allows the adversary to generate false detections in distant regions. Nonetheless, we show that false negative failures can be circumvented by applying feature denoising and adversarial training. However, we observe that distant false positives are much harder to correct with adversarial defense, as they are also caused by inaccurate mappings between 2D pixels and 3D LiDAR points during fusion.

## 2 Related work

Adversarial attacks were first discovered in the 2D image domain, where small perturbations on the pixels were shown to generate drastically different prediction results on object classification [19, 20]. Networks trained on object detection and semantic segmentation have also been shown to exhibit such vulnerability [8, 9, 21, 22, 23, 24]. Early methods [19, 20, 25, 26] assume the perfect knowledge of the gradients of the victim model, referred to as *whitebox* attacks. Later it was found that a *blackbox* attack can achieve similar success as well [27, 28, 29]. Defense and robustness evaluation procedures have also been explored for adversarial attacks [30, 31, 32, 33, 34, 35, 36, 37].

Aside from changing the pixel values by a small amount, various other ways to "perturb" an image were also proposed. Object insertion attacks are realistic attacks that insert an object to change the network output while not introducing changes in semantics [38, 15, 7, 39, 40]. These attacks were originally designed to be stickers that can be attached to a target object, and has since also been applied to the lens of a camera [41]. Image rendering is another popular technique for non-pixel based attacks, which can also be made differentiable [42], by using which [43] showed that adversarial attacks can be made through changing lighting. Various other object insertion attacks designed camouflage textures that can be wrapped around the target object [44, 10, 45, 46, 47, 48].

Aside from the typical image-based attacks introduced above, adversarial attacks against point clouds have also been studied. [17, 49, 50, 51] tried to directly perturb the location and cardinality of point clouds. However, such attacks may not be physically realizable, as arbitrary perturbations won't be achievable by a LiDAR sensor with fixed angular frequency. Towards more realistic attacks, [52, 53] developed spoofing attacks that add malicious LiDAR points, while other

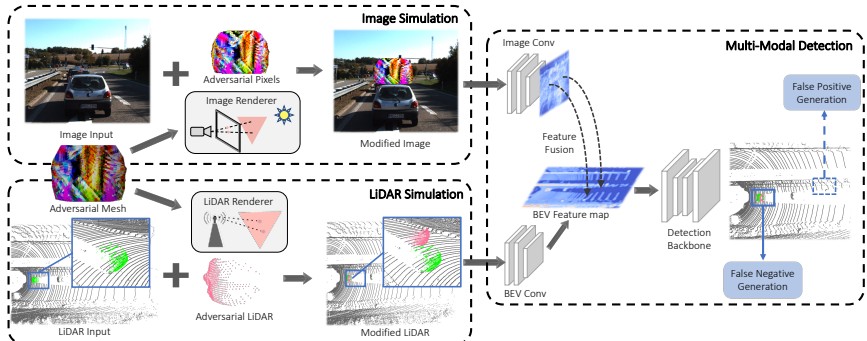

Figure 2: Overview of the attack pipeline. The adversarial mesh is rendered into both LiDAR and image inputs in a differentiable manner. The inputs are then processed by a multi-sensor detection model which outputs bounding box proposals. An adversarial loss is then applied to generate false negatives by suppressing correct proposals and false positives by encouraging false proposals. Since the entire pipeline is differentiable, gradient can flow from the adversarial loss to mesh parameters.

approaches [16, 54, 12] instead optimize 3D mesh surfaces and use differentiable ray-casting to generate LiDAR point clouds.

Despite the fact that multi-modal sensor configurations are widely seen on self-driving vehicles [55, 5, 56, 57, 58, 59], research on multi-modal sensor attacks is still very limited. Several preliminary works show the possibility of attacking multi-sensor fusion networks [13, 14, 60]. However, [13] did not consider consistency across data modalities when perturbing the image input, whereas [14] did not consider image texture, resulting in a lack of attack expressivity, and [60] did neither. We believe that it would be an interesting question to ask, whether multi-sensor fusion can be made more robust when the attacks are both input-agnostic and physically realizable.

## 3   Multi-sensor Adversarial Learning

In this section, we present a general method for learning an adversarial textured mesh to attack any multi-sensor object detector that is differentiable end-to-end. Specifically, we require the adversarial attack to be (1) input-agnostic for different environments, (2) geometrically-consistent across image and LiDAR input modalities, and (3) fully-automatic for implementation at large-scale. Our attacks are focused on vehicles as they are the most common object of interest on the road.

**Preliminaries:** We consider a bird's eye view (BEV) object detection model $F$ that takes the front camera image $x_I \in [0,1]^{H \times W \times 3}$ and LiDAR point clouds $x_L \in \mathbb{R}^{P \times 3}$ as input $x = (x_I, x_L)$. Here, the dimensions $H$ and $W$ represent the image height and width respectively. $P$ is the number of LiDAR points which could vary in each frame. The object detector is trained on bird's eye view (BEV) bounding box annotations $\mathcal{Y}$, with each bounding box instance $b \in \mathcal{Y}$ parameterized by $b = (b_x, b_y, b_h, b_w, b_\alpha)$. Subsequently, $b_x$ and $b_y$ are coordinates of the bounding box center, $b_h$ and $b_w$ indicate the width and height, respectively, and $b_\alpha$ represents the orientation. In order to process both image and LiDAR data modalities, the object detector uses two separate branches to extract features from each modality (see Fig 2). Then, the 2D image features are projected into 3D space to be fused with the LiDAR features.

### 3.1   Multi-sensor Simulation for Object Insertion

In this work, we design a framework to insert a textured mesh into the scene so that both appearance and shape can be perturbed to attack multi-sensor perception systems. We attach a triangle mesh $\mathcal{M} = (\mathcal{V}, \mathcal{F}, \mathcal{T})$ onto the roof of a host vehicle, as such placement is physically realizable in the real world. The mesh is parameterized by vertex coordinates $\mathcal{V} \in \mathbb{R}^{N \times 3}$, vertex indices of faces $\mathcal{F} \in \mathbb{N}^{M \times 3}$, and per-face vertex textures $\mathcal{T} \in \mathbb{R}^{M \times C \times C \times 3}$. The dimensions $N$, $M$ and $C$ represent the number of vertices, the number of triangle faces, and the per-face texture resolution, respectively. For scalability reasons, we do not consider transparency, reflective materials, or shadowing, as handling each case would require sophisticated physics-based rendering. Instead, we approximate the sensor simulation using LiDAR ray-tracing and a light-weight differentiable image renderer. Both image and LiDAR rendering pipelines are differentiable, allowing gradients from LiDAR points and

image pixels to flow into the mesh parameters during optimization. The overall pipeline of for object insertion is illustrated in Figure 1.

**Rooftop Approximation:** First, we estimate the center of the vehicle's rooftop to determine the 3D location for placing the adversary. Following [61, 12, 62], we represent our vehicle objects using signed distance functions (SDFs) and further project them onto a low-dimensional shape manifold using PCA. For each vehicle, we then optimize the low-dimension latent code that minimizes the fitting error between the vehicle point clouds and the shape manifold. Then, a fitted SDF is converted to a watertight vehicle mesh with Marching Cubes [63]. We select the top 20cm of the vehicle as the approximate rooftop and use the rooftop center and vehicle heading to determine the exact pose for object insertion.

**LiDAR Simulation:** To simulate insertion in the LiDAR sweep, we sample rays according to Li-DAR sensor specifications used to collect the original sweep, such as the number of beams and the horizontal rotation rate. We then compute the intersection of these rays and mesh faces using the Moller-Trumbore algorithm [64, 12] to obtain a simulated point cloud of the adversarial mesh. These simulated points are then added to the original LiDAR sweep.

**Image Simulation:** To render the adversary into the image, we extract the intrinsics and extrinsics from the camera sensor that captured the original image. We then use a light-weight differentiable renderer SoftRas [65] to simulate the mesh pixels. Using a soft rasterizer during optimization allows gradient flow from occluded and far-range vertices to enable better 3D reasoning from pixel gradients. To enhance the fidelity of rendered images, we model the sun light with a directional lighting model as a light at infinite distance.

**Occlusion Reasoning:** As we insert a new object into the scene, the rendering process must also consider occlusion for both the original and newly rendered points and pixels. To handle LiDAR occlusions, we compare the depth of existing points in the LiDAR sweep and newly rendered points, discarding the point that is farther away. Unlike LiDAR where the depth of each point is known, raw images do not contain depth information. To obtain depth estimates, first the LiDAR points are projected onto the image to generate a sparse depth image since images have higher resolution than LiDAR. We then use a depth completion model [66], which uses the sparse depths and RGB image to compute dense per-pixel depth estimates. With depth estimates of each pixel, we overlay rendered pixels onto the original image and discard occluded pixels. Note that we do not attack the depth completion model as it is a preprocessing step used to increase the fidelity of simulation.

## 3.2 Universal Adversarial Example Generation

**Adversarial Objectives:** We consider two adversarial objectives: one for false negatives and the other for false positives. To generate false negative attacks, we follow prior work [8, 12] in attacking object detectors by suppressing all relevant bounding box proposals for the host vehicle. A proposal is relevant if its confidence score is greater than 0.1 and it overlaps with the ground-truth bounding box. The adversarial loss then minimizes the confidence of all candidates:

$$\mathcal{L}_{\text{adv}}^{\text{fn}} = \sum_{b \in \widetilde{\mathcal{Y}}} -\text{IoU}(b, b^*) \log(1 - \texttt{score}(b)), \tag{1}$$

where IoU denotes the intersection over union operator and $b^*$ is the corresponding ground-truth box we aim to attack.

Alternatively, we aim to generate false bounding box proposals that do not overlap with any ground-truth boxes in the scene. The false positive adversarial loss increases the confidence of the false positive candidates as follows:

$$\mathcal{L}_{\text{adv}}^{\text{fp}} = \Sigma_{b \in \widetilde{\mathcal{Y}}_{\text{fp}}} \log(1 - \texttt{score}(b)) \text{ and}$$
$$\widetilde{\mathcal{Y}}_{\text{fp}} = \{b | b \in \widetilde{\mathcal{Y}} \text{ and } \forall b^* \in \mathcal{Y} \text{ s.t. IoU}(b, b^*) = 0\}, \tag{2}$$

where $\widetilde{\mathcal{Y}}_{\text{fp}}$ is a subset of bounding box proposals with no overlap with any ground-truth object bounding boxes.

**Mesh Regularization:** Besides the adversarial objective, we use an additional regularization term to encourage realism in the mesh geometry. Specifically, we use a mesh Laplacian regularizer [65], which encourages smooth object surface geometries: $\mathcal{L}_{\text{lap}} = \sum_i \|\delta_i\|_2^2$, with $\delta_i$ as the distance from

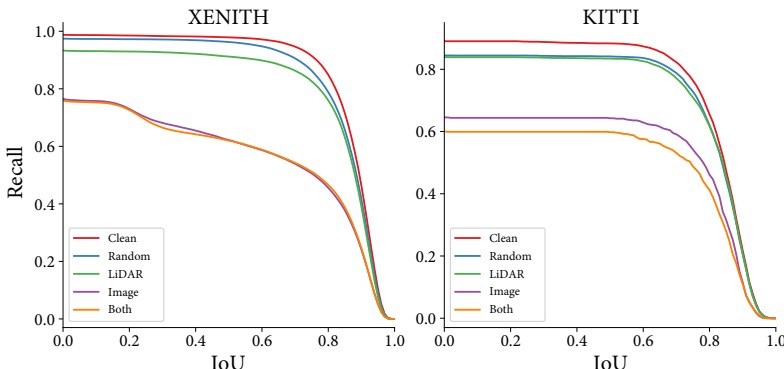

Figure 3: Plot of the host vehicle recall across IoU thresholds. Only attacking LiDAR yields very weak attacks and attacking the image produces significantly stronger perturbations.

vertex $v_i \in \mathcal{V}$ to the centroid of its immediate neighbors $\mathcal{N}(i)$: $\delta_i = v_i - \frac{1}{\|\mathcal{N}(i)\|} \sum_{j \in N(i)} v_j$. In addition to Laplacian regularization, we also constrain the physical scale of the adversary with an axis-aligned 3D box. Namely, we require that $\|\mathcal{V}_j\|_\infty \leq L_j$ for $j \in \{x, y, z\}$, where $L_x$, $L_y$, and $L_z$ represent the box constraints along $xyz$-axis, respectively.

**Learning Input-Agnostic Attacks:** Overall, our optimization objective can be summarized as $\mathcal{L} = \mathcal{L}_{\text{adv}}^{\text{fn}} + \lambda_{\text{fp}} \mathcal{L}_{\text{adv}}^{\text{fp}} + \lambda_{\text{lap}} \mathcal{L}_{\text{lap}}$, where $\lambda_{\text{fp}}$ and $\lambda_{\text{lap}}$ are coefficient that weight the relative importance of the false positive loss term and mesh regularization term. We employ this objective to optimize the shape and appearance of the inserted object on the entire dataset to generate an input-agnostic adversarial example. Therefore, we can denote the optimal adversary as the following equation:

$$\mathcal{M}^* = \underset{\mathcal{M}}{\text{argmin}} \; \underset{x, \mathcal{Y}}{\mathbb{E}} \left[ \mathcal{L}_{\text{adv}}^{\text{fn}} + \lambda_{\text{fp}} \mathcal{L}_{\text{adv}}^{\text{fp}} + \lambda_{\text{lap}} \mathcal{L}_{\text{lap}} \right]. \tag{3}$$

With our proposed pipeline which is differentiable end-to-end, optimization of the adversarial mesh is done using *projected gradient descent* to respect the $\ell_\infty$ constraints on mesh vertices. In our experiments, we also conduct attacks targeting a single modality. To achieve this, we disable the gradient flow to the untargeted input branch, while we still simulate the mesh into both modalities to maintain physical consistency across image and LiDAR modalities.

### 3.3 Multi-sensor Adversarial Robustness

Towards defenses against our object insertion attack, we also study defense mechanisms. Compared to the single-sensor setting, achieving multi-sensor adversarial robustness is even more challenging. First, each single input modality could be attacked even when the perturbations on the other input sensors are non-adversarial. Second, adversarial perturbations from each single input modality can interact with each other, which is a unique aspect in the multi-sensor setting. Thus, we need to deal with not only perturbations at each input modality but also their effect in the fusion layer.

We employ adversarial training as it is the most standard and reliable approach to defense. Adversarial training can be formulated as solving for model parameters

$$\theta^* = \underset{\theta}{\text{argmin}} \; \underset{x, \mathcal{Y}}{\mathbb{E}} \left[ \underset{\widetilde{x}_\text{I}, \widetilde{x}_\text{L}}{\max} \mathcal{L}_{\text{task}}(F(\widetilde{x}; \theta), \mathcal{Y}) \right] \tag{4}$$

which minimize the empirical risk under perturbation. Here $\mathcal{L}_{\text{task}}$ is the loss function used to train the detection model. This is achieved by training detection model $F$ against adversarial data $\widetilde{x}$ generated by our attack. While adversarial training is typically performed on image perturbations that are cheap to generate with only few PGD steps [67], our adversarial example generation is prohibitively expensive for the inner loop of the min-max objective. Thus, instead of generating a strong adversary from scratch at every iteration, we adopt free adversarial training [68] and continuously update the same adversary to reduce computation.

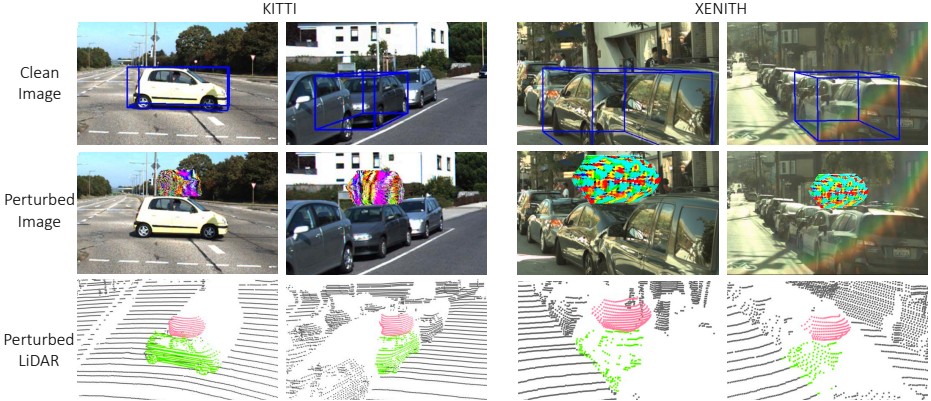

Figure 4: Placing the adversarial mesh on a host vehicle can hide the host vehicle completely from state-of-the-art detectors.The same mesh is used for all vehicles in a dataset as the attack is input-agnostic with respect to the training distribution.

## 4 Experimental Evaluations

In this section, we first describe the our datasets, attack protocols, and evaluation metrics. More details on experiment setting are provided in the supplementary material. We then conduct experiments on the multi-sensor detection model MMF [5] and present our empirical findings for *white-box* attacks on each dataset and the *black-box* transfer attacks across datasets. Finally, we explore several defense mechanisms to achieve a more robust multi-sensor object detector.

### 4.1 Experimental Setting

**Datasets:** We conduct our experiments on two self-driving datasets: KITTI [18] and XENITH. XENITH is collected by having a fleet of self-driving cars drive around cities in North America. Snippets are captured in the daytime and detection is performed on objects within 70 meters forward and 40 meters to the left and right of the ego-car. Each vehicle in a frame is considered a separate sample and we have 12,284 vehicles in KITTI and 77,818 in XENITH.

**Metrics:** Following prior work on object insertion attacks [12], we evaluate how often the host vehicle "disappears" by measuring its *recall* across various IoU thresholds. For a scalar metric, we evaluate the *false negative attack success rate (FN ASR)* as the percentage of host vehicles detected before perturbation that are undetected after perturbation. We consider a vehicle detected if there exists an output bounding box having greater than 0.7 IoU with the vehicle. On the other hand, we consider an output bounding box a false positive if its maximum IoU with any ground truth box is less 0.3 and it does not overlap with any detection produced before perturbation. We evaluate the *false positive attack success rate (FP ASR)* as the percentage of attacks which generate at least one false positive. Finally, the overall *attack success rate (ASR)* is the percentage of attacks which successfully create a false positive or false negative.

**Implementation Details:** The adversarial mesh is initialized as an icosphere with $N = 162$ vertices and $M = 320$ faces. Per-face textures are parameterized using a texture atlas with a $5 \times 5$ texture resolution for each face. During optimization, we set $\lambda_{\text{lap}} = 0.001$, $\lambda_{\text{fp}} = 1$, and further constrain the scale of the mesh with an axis-aligned 3D box where the $x$ and $y$ coordinates are bounded by $L_x = L_y = 0.8m$ and the $z$ coordinate is bounded by $L_z = 0.5m$. We use Adam [69] to optimize the mesh parameters with a learning rate of $0.004$ for textures and $0.001$ for vertex coordinates. To target either LiDAR or image branch in isolation, we disable gradient flow to the other branch during the backward pass to the adversary.

### 4.2 Universal Adversarial Attacks

**Hiding Host Vehicle:** We evaluate the drop in recall in detecting the host vehicle, as missed detections can lead to colliding with unseen objects which is the most dangerous outcome. We sweep IoU thresholds and visualize the IoU-recall curve in Figure 3. First, inserting a mesh with random-

| Detector | Attack | FN ASR | FP ASR | ASR |
|---|---|---|---|---|
| LiDAR | LiDAR [12] | 31.85% | 4.84% | 33.23% |
| LiDAR + Image | Random | 5.68% | 2.01% | 7.64% |
| | LiDAR | 7.99% | 2.36% | 10.11% |
| | Image | 26.06% | 3.40% | 28.43% |
| | Both | **32.76%** | **4.38%** | **34.68%** |

Table 1: Attacks on KITTI. Also comparing with random meshes and a LiDAR only model.

| Detector | Attack | FN ASR | FP ASR | ASR |
|---|---|---|---|---|
| LiDAR | LiDAR [12] | 23.80% | 10.70% | 32.60% |
| LiDAR + Image | Random | 5.06% | 4.15% | 9.17% |
| | LiDAR | 9.52% | 6.21% | 15.33% |
| | Image | 42.81% | 10.78% | 49.59% |
| | Both | **43.15%** | **11.77%** | **49.76%** |

Table 2: Attacks on XENITH. Also comparing with random meshes and a LiDAR only model.

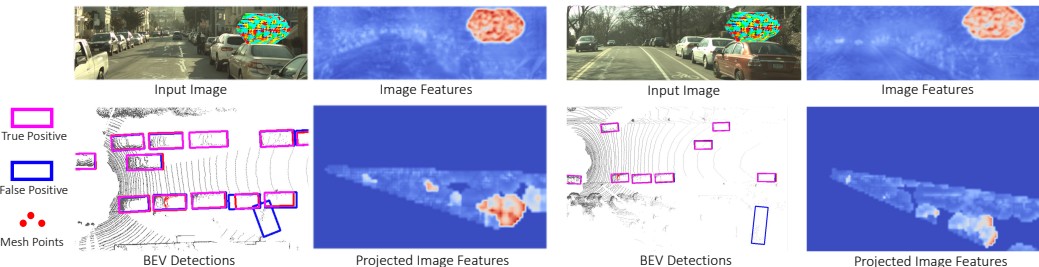

Figure 6: Visualization of the attack producing distant false positives due to the camera's perspective projection, as well as the corrupted image features in the image plane and projected in 3D.

ized shape and appearance has little impact on the detector. On the other hand, an adversarial mesh generated by perturbing both input modalities leads to a significant drop in recall. Moreover, we perturb the LiDAR and image inputs in isolation and find that targeting the LiDAR inputs alone yields very weak attacks. Meanwhile, targeting the image alone is almost as strong as perturbing both modalities. Therefore, image inputs are significantly less robust to the proposed attack.

**Attack Success Rates:** In Table 1 and Table 2, we further analyze results in terms of the attack success rates. Again, we consider meshes with randomly generated geometry and texture as a baseline. We observe similar trends of image features being significantly more vulnerable. In addition to missed detections, the adversarial mesh is able to attack the detector through generating false positive proposals of objects that do not exist. Furthermore, we compare against prior work [12] which attacks a LiDAR-only detector. In this case, incorporating image inputs boosts robustness to LiDAR attacks at the cost of being more vulnerable to multimodal attacks.

**Adversary Size** Furthermore, to understand how much the size of the adversary affects the strength of the attack, we vary the size of the box constraints on the vertex parameters. Here we sweep $L_x = L_y = L_z = L$ and and measure the attack success rates. Results are shown in Figure 5. As expected, the attack becomes stronger as the $\ell_\infty$ constraints on vertex coordinates are relaxed.

Figure 5: Size of adversary box constraint vs attack success rate.

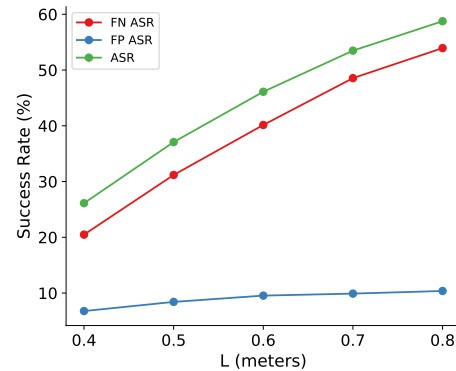

**Qualitative Examples:** Qualitative examples are shown in Figure 4. First, the detector fails to detect host vehicles with the adversarial mesh on its rooftop. We show detections in the image rather than LiDAR for ease of viewing. Note that the same adversarial mesh is used for all demonstrations, as the attack is agnostic to the host vehicle and environment. Furthermore, we show in Figure 6 that our adversarial mesh generates false positives at very distant locations. Here, detections are visualized in BEV since distant objects appear too small in the image. Additionally, we visualize image features in the image plane and the visual cone of projected image features into 3D, showing that long-range false positives are caused by strong image features dominating after fusion.

**Black-box Transfer Attacks:** We conduct transfer attacks across datasets and show results in Table 3. Overall, our transfer attack on the target dataset is stronger than attacking only the LiDAR

| Source | Target | FN ASR | FP ASR | ASR |
|---|---|---|---|---|
| KITTI | KITTI | 32.76% | 4.38% | 34.68% |
| | XENITH | 14.20% | 2.86% | 16.88% |
| XENITH | KITTI | 12.64% | 6.12% | 18.22% |
| | XENITH | 43.15% | 11.77% | 49.76% |

Table 3: Transfer - KITTI & XENITH

| Defense | FN ASR | FP ASR | ASR | AP(clean) |
|---|---|---|---|---|
| None | 43.15% | 11.77% | 49.76% | **84.64%** |
| JPEG [70] | 43.19% | 9.45% | 49.60% | 84.52% |
| Adv Train [68] | 7.83% | 8.29% | 14.97% | 84.16% |
| Adv FD [36] | **3.57%** | **7.53%** | **10.82%** | 83.97% |

Table 4: Defense results on XENITH

input modality on the source dataset, especially from XENITH to KITTI. On the other hand, the transferability is probably lowered by the image resolution and hardware, which is beyond the scope of our paper but an interesting future direction to explore.

### 4.3 Improving Robustness

**Attacks Against Defense Methods:** As empirical findings suggest that the image feature is more vulnerable, we first employ an existing image-based defense method that removes high-frequency component through JPEG compression [70]. In addition, we conduct adversarial training against the attacker. Since generating a strong adversary is extremely expensive due to the simulation pipeline, we employ a strategy similar to Free Adversarial Training [68] and reuse past perturbations by continuously updating the same adversarial object. Specifically, we perform 5 updates to the adversary per one update to the model. We combine the feature denoising [36] with the adversarial training to further enhance robustness against image perturbations in particular. We report the success rates as well as the average precision (AP) at 0.7 IoU to study the trade-off between adversarial robustness and performance on benign data [35].

As shown in Table 4, we find JPEG compression is very ineffective as defense. We hypothesize this is because the input-agnostic adversary is rendered at various different poses during training and therefore do not rely on high-frequency signals that are removed by JPEG compression. In comparison, our adversarial training effectively reduces the overall attack success rate from 49.76% to 14.97%, while dropping AP by 0.5%. Finally, adding non-local mean blocks after every residual block in the image processing backbone further improves robustness by reducing the ASR by 5%.

**Discussions and Future Work:** While adversarial training methods are effective, they are specific to a specific threat model and may not generalize to unseen perturbations. A more threat-agnostic mechanism like more robust sensor fusion would bring more robustness in general. Furthermore, adversarial defense methods are only effective at recovering the missed detections, but struggle to detect false positives. We believe this is because distant false positives shown in Figure 6 are only partially due to vulnerabilities to adversarial perturbations. In fact, such examples exploit erroneous associations between objects that are distant in 3D. Specifically, the mapping between a mesh pixel and a LiDAR point far away from the mesh enables such attacks. These false associations can easily occur if the assigned pixel for each LiDAR points is shifted by a few pixels, since objects which are far apart in 3D may appear very close in 2D. We identify two reasons how this can occur in practice. First, due to the receptive field of DNN activations, an adversarial object can influence pixels outside its physical boundaries. Second, while LiDAR sweeps are collected in a continuous fashion with a rotating sensor, images are captured instantaneously at regular intervals. Consequently, the camera extrinsics used for projection become outdated for LIDAR points captured before and after the image. Thus, to achieve more robust sensor fusion for images and LiDAR, fusion modules must reason about 3D geometry, contextual information, and temporal information of LiDAR points to generate mappings between image pixels and LiDAR points more intelligently. We hope these findings will inspire future work towards more robust sensor fusion methods.

## 5 Conclusions

Our work investigates practical adversarial attack against mulit-sensor detection models in self-driving to understand how consuming multiple input modalities affect adversarial robustness. Compared to existing attacks against multimodal detectors, our object insertion attack is more threatening in practice as we generate input-agnostic and physically realizable adversarial perturbations. Our experiments reveal that vulnerabilities of multi-sensor object detectors are primarily due to non-robust image features. While adversarial training can effectively recover the missed detections, we find it still struggles to detect false positives without a deeper reasoning about 3D geometry in feature fusion. We believe this work would open up new research opportunities and challenges in the field of multi-sensor robustness.

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
