# OpenReview forum: "Exploring Adversarial Robustness of Multi-sensor Perception Systems in Self Driving"
_robot-learning.org/CoRL/2021/Conference — CoRL2021 Poster_

### Official Review · Reviewer_2ySC · 2021-07-04

**Originality:** Good
**Technical Quality:** Good
**Clarity Of Presentation:** Very Good
**Impact:** 4

**Recommendation:**

Weak Accept: I recommend accepting the paper, but will not argue for my recommendation if the majority of other reviewers have a different opinion.

**Summary:**

This paper presents an adversarial perturbation method to multi-modal inputs (e.g., LIDAR point cloud & image) showcase the susceptibilities of multi-sensor object detection systems on self-driving cars. Different from existing works that only investigate proof-of-concept adversarial methods on image data via perturbation with imperceptible noises (whose physical realizability is questionable),  the presented method focuses on physically realizable attacks via (adversarial) object insertion. In this approach, the adversary is represented via a parametrized texture mesh, which can be optimized and rendered into an object to be mounted on top of self-driving cars.

This is achieved by inserting the mesh into the LIDAR point cloud and project it onto the original 2D image, from which pixel depths can be estimated via an off-the-shelf depth estimation algorithm. These results in parameterized LIDAR & image modifications (their defining parameters are those of the texture mesh) which can be optimized to maximize the rate of false positive & false negative over the detection algorithm's bounding box proposal. The optimized texture mesh is then treated as an input-agnostic adversary that can be inserted in any (test) images & corresponding LIDAR point cloud. The effectiveness of the proposed method is also studied empirically in a transfer setting where it is computed on one dataset (e.g., KITTI) but applied & tested on another (e.g., XENITH). All reported results show promising & interesting results.

**Issues:**

Please refer to my specific comments & questions in the weakness section above.

**Reviewer Expertise:**

Good: General knowledge of the area

**Strengths And Weaknesses:**

STRENGTH:

The method is well-motivated from a very practical angle. Instead of simply demonstrating a proof-of-concept via crafting imperceptible noises to perturbs images adversarially (which might not be realizable physically), the proposed method studies a new method that aims to render a physical object with adversarial texture that can be naturally inserted into the scene to perturb both the LIDAR & image input to the object detection system on self-driving car.

The paper is very well-written & the experiments are also well-presented, which sufficiently demonstrate the effectiveness of the proposed method.

From the proposed attack, an adversarial training algorithm is also presented which helps decrease the success rate of the attack, thus making this a completed & motivating research.

WEAKNESSES:

I noticed that the trade-off coefficient of the Laplacian regulariser is rather small in the experiment, which apparently makes the size of the mesh relatively large (as shown in many visual excerpts). This unfortunately makes the adversary less realistic & realizable -- while the attack is effective, the perturbed images do not look natural at all.

Could the authors elaborate more on the effectiveness of the method with varying size of the texture mesh? In addition, I think it would also be important to compare with image-based attack methods (e.g., simply put a sticker on the host vehicle rather than mount a 3D object on top of it) to highlight the advantage of being able to create an attack on multiple sensor modalities.

Furthermore, the attack method has been evaluated against specific-defense methods but it remains unclear how effective it is against black-box defense methods? It would be great if the authors could comment on whether such experiment is possible?

Another (more minor) point: For thoroughness, please consider running the evaluation on multiple, independent data split to report the confidence of the success rate estimation.

**Summary Of Recommendation:**

In general, I like the proposed idea, especially its practical angle on a physically realizable attack via natural object insertion. That being said, I feel that the evaluation here is still pretty much artificial at this point but I acknowledge that this is an interesting concept that might inspire more follow-up investigations. Given this, my rating for this work is weak accept.

---

> ### Author Response · Authors · 2021-08-25
> **Author Response To Reviewer 2ySC**
>
> Thank you for your review and we are glad you appreciate the practical approach and specifically our emphasis on physically realizable attacks.
>
> `R3-Q1` - Effectiveness of method with varying sizes of texture mesh
>
> First, we would like to point out that the Laplacian regularizer only regularizes the smoothness of the mesh and does not have a significant impact on size. Rather, the size is controlled by the box constraint parameters [L_x, L_y, L_z] (see section 4.1). To elaborate more on the effectiveness of the method with varying mesh sizes, we have moved an experiment from the supplementary material into the main manuscript in Figure 5. Here, we sweep L = L_x = L_y = L_z and measure attack success rates and show how the attacker becomes weaker with smaller meshes and stronger with bigger meshes.
>
> `R3-Q2` - Comparison with image-based attack methods
>
> Regarding comparisons with image attacks, our experiments in Table 1 & 2 demonstrate only attacking the image modality versus only attacking the LiDAR or attacking both. We believe variations of the same threat model provides a more fair comparison of the input modalities versus experiments on two different threat models. Thanks for the suggestion on using adversarial stickers as it is a very practical attack against image inputs. However, we designed our experiments to vary input modalities within the same threat model to provide a clear understanding of how each input mode affects robustness. These experiments show that attacking LiDAR only is not very effective while attacking image only is almost as effective as attacking both. Also, we see that the multi-sensor model is more vulnerable than a LiDAR-only model. From this, we can understand how much stronger attacks get when they also perturb the image modality.
>
> `R3-Q3` Black box defense methods
>
> While adversarial training and feature denoising are defense methods are specific to the detector and threat model, we would like to point out that our experiment in Table 4 also considers applying JPEG compression and decompression to image inputs which is a black box defense method. However, we found this method to be ineffective. Towards more general methods for robustness, we plan to investigate more robust sensor fusion algorithms. In particular, our discussion in Section 4.3 identifies vulnerabilities caused by incorrect associations between image pixels and LiDAR points which are difficult to correct via adversarial defenses and instead require more robust sensor fusion modules. General improvements on sensor fusion robustness can be thought of as a black box defense method since it's agnostic to the threat model.
>
> `R3-Q4` - Evaluation on multiple data splits
>
> Agreed, thank you for the suggestion on how to improve our experiments.

---

### Official Review · Reviewer_4ZtU · 2021-07-07

**Originality:** Fair
**Technical Quality:** Fair
**Clarity Of Presentation:** Very Good
**Impact:** 3

**Recommendation:**

Weak Accept: I recommend accepting the paper, but will not argue for my recommendation if the majority of other reviewers have a different opinion.

**Summary:**

This paper addresses the problem of assessing the robustness of
multi-sensor neural networks, as used in robotic applications. The
authors develop a physical-like attack (by simulating an object on the
vehicle in front) such that the attack affects both measurement
modalities (namely, images and lidar) in a consistent way. The authors
provide an evaluation over two datasets, which suggests that using
multi-sensor networks does improve robustness to individual-sensor
attacks but may be susceptible to multi-sensor attacks.

**Issues:**

N/A

**Reviewer Expertise:**

Excellent: Expert knowledge on the topic of the paper

**Strengths And Weaknesses:**

The paper is well written, and the approach is clear. I like the idea
of using sensor fusion to improve robustness to adversarial attacks.

My first comment has to do with the paper's contribution. The authors
claim that their main contribution is developing an attack that is
consistent across different sensor modalities. However, no
quantitative evaluation of the proposed attack generation is
provided. How can one tell (quantitatively) that the proposed attack
is physically consistent across sensors? A real-world example(s) would
also help demonstrate that this attack choice is valid.

My second comment is that I think the sensor fusion aspect is not
sufficiently explored. In particular, it seems that the utilized
sensor fusion method is putting too much weight on the image
processing part, which is why even image attacks on their own are
quite successful. Instead of retraining the detectors using
adversarial training, I think that coming up with a better sensor
fusion approach would bring more robustness in general. The reason for
this is that adversarial training is performed for the specific attack
implemented in this paper. On the other hand, a robust sensor fusion
approach would make sure that the physical attack needs to be very
precisely coordinated in order to modify the various sensor modalities
in an adversarial way.

**Summary Of Recommendation:**

Overall, I think that using sensor fusion to improve robustness is a promising research direction and should be explored further. Although I think the paper could provide better sensor fusion analysis, I think it is a good first step in this field.

---

> ### Author Response · Authors · 2021-08-25
> **Author Responses To Reviewer 4ZtU**
>
> We thank the reviewer for her/his comments on how to improve the paper.
>
> `R2-Q1` - Quantitatively check the proposed attack is physically consistent across sensors.
>
> First, we would like to point out that the perturbations in image and LiDAR are consistent by construction since they are both captured from the same underlying mesh representation. Since small mismatches may still occur if there are rendering errors from pose noises or low fidelity renderers, one way to quantitatively check consistency between the inputs is to compare the camera image with the range-view image of the LiDAR sweep. Here, we would convert the images to have the same resolution and take the mask of the adversary in both images and compute their intersection-over-union. However, the consistency between these two masks is checked implicitly when we handle occlusion, as the mask of the adversary in the RGB image is actually determined by the range-view image of the LiDAR.
>
> `R2-Q2` - More exploration on sensor fusion
>
> As a result of our experiments and analysis on the proposed adversarial attack, we identified that sensor fusion modules may suffer from incorrect associations between image pixels and LiDAR points and discuss the reasons in detail in section 4.3 of the manuscript. We employ adversarial training as it’s a common first step towards understanding how well the vulnerabilities can be corrected. This is what led us to find the sensor fusion errors which are difficult to correct via adversarial defense. Thus, we agree that improved sensor fusion is a more general robustness mechanism that is agnostic to the threat model and have revised the manuscript to include this discussion point in section 4.3. Overall, more robust sensor fusion is definitely an interesting direction to explore and we are glad you recognize our work as a good first step in this field.

---

> ### Comment · Reviewer_4ZtU · 2021-08-31
> **Post-Rebuttal Comment**
>
> I appreciate the effort the authors put in during the rebuttal phase. I think this is a solid paper, and I am in favor of acceptance.

---

### Official Review · Reviewer_2uDN · 2021-07-23

**Originality:** Good
**Technical Quality:** Very Good
**Clarity Of Presentation:** Very Good
**Impact:** 4

**Recommendation:**

Weak Accept: I recommend accepting the paper, but will not argue for my recommendation if the majority of other reviewers have a different opinion.

**Summary:**

The paper presents a framework for having multi-modal adversarial attacks in self-driving domain, and uses both pixel images and LiDAR attacks. The authors perform extensive experiments grounded in practical self-driving settings, and proposes consistent perturbations among the input modalities.

**Issues:**

It would be great if the authors could respond to some of the concerns listed in the Strengths and Weaknesses section.

**Reviewer Expertise:**

Fair: Some knowledge of the area

**Strengths And Weaknesses:**

Strengths:

The paper made a clear contribution to the study of multi-modal adversarial attacks in self-driving domain. The design of the experiments are technically sound, and the authors gave a clear and extensive literature review of the prior works and introduced the method in this paper pretty well.

Weaknesses (or some thoughts upon the paper):

One thought I'm having is whether it's worth considering more subtle adversarial attacks on top of the adversarial mesh. For example, in image classification adversarial attacks work there has been minor changes on road signs, changing colors of certain parts of the images etc that are semantically meaningful to human's discretion. I wonder if the same thing could be done to the images and LiDAR inputs in the self-driving setting.

Also, is it possible to have some qualitative analysis of the result of detecting adversarial attacks? For example, it could show some success / failure examples of different methods and therefore shows how this method outperform the previous ones in a qualitative way.

One other minor point is potentially some implementation details could go into the appendix, and the main paper could be spent on more discussion / analysis.

**Summary Of Recommendation:**

I recommend the paper for acceptance as multi-modal adversarial attacks is an interesting yer understudied problem in self-driving domain, and the authors has demonstrated success in attacks in multi-model inputs.

---

> ### Author Response · Authors · 2021-08-25
> **Author Responses To Reviewer 2uDN**
>
> Thank you for the review, we are glad you found the design of the experiments technically sound. We have reflected your suggestion on moving implementation details into the supplementary material and brought another experiment into the main manuscript instead.
>
> `R1-Q1` - More subtle attacks on top of the adversarial mesh
>
> Thank you for the suggestion! Our work focuses on physically realizable multi-modal attacks to understand how consuming multiple input modalities affects the adversarial robustness of detection models. Our main purpose is to analyze if some modalities are more vulnerable than others and how this can affect overall robustness. Therefore, our experiments are limited to a single threat model where we can vary which modalities are perturbed. While introducing subtle perturbations in addition to the current attack is a great suggestion for improving the strength of the attack, we believe it falls outside the scope of this work which aims to understand how processing multi-sensory inputs affects adversarial robustness.
>
> `R1-Q2` - Qualitative analysis of the proposed method versus previous methods
>
> Figure 6 of the revised manuscript visually demonstrates a type of attack resulting from projecting 2D image features into 3D. To the best of our knowledge, distant false positives resulting from this projection is a novel result specific to attacks on multi-sensor fusion models.

---

> > ### Comment · Reviewer_2uDN · 2021-09-03
> > **Thank you for your response**
> >
> > Thank you the authors for the effort put in the rebuttal period! I'm in favor of acceptance like my original review.

---

### Meta-Review · Area_Chair_K7rQ · 2021-08-13

**Recommendation:** Accept (Poster)
**Confidence:** 4

**Metareview:**

This paper investigates the problem of adversarial robustness of multi-sensor perception systems, with a focus on the autonomous driving domain. While the reviewers are generally positive about this paper, they also raise a number of important concerns. The authors should carefully address the reviewers' comments in their rebuttal.

UPDATE POST DISCUSSION PHASE: I would like to thank the authors for their comments and clarifications during the discussion phase. The reviewers agree that this paper provides a valuable contribution, and I concur with this assessment.

---

### Decision · Program_Chairs · 2021-09-13

**Decision:**

Accept (Poster)

**Comment:**

This paper investigates the problem of adversarial robustness of multi-sensor perception systems, with a focus on the autonomous driving domain. While the reviewers are generally positive about this paper, they also raise a number of important concerns. The authors should carefully address the reviewers' comments in their rebuttal.

UPDATE POST DISCUSSION PHASE: I would like to thank the authors for their comments and clarifications during the discussion phase. The reviewers agree that this paper provides a valuable contribution, and I concur with this assessment.